# PASSION: Towards Effective Incomplete Multi-Modal Medical Image Segmentation with Imbalanced Missing Rates

## ABSTRACT

Incomplete multi-modal image segmentation is a fundamental task in medical imaging to refine deployment efficiency when only partial modalities are available. However, the common practice that complete-modality data is visible during model training is far from realistic, as modalities can have imbalanced missing rates in clinical scenarios. In this paper, we, for the first time, formulate such a challenging setting and propose Preference-Aware Self-diStillatION (PASSION) for incomplete multi-modal medical image segmentation under imbalanced missing rates. Specifically, we first construct pixel-wise and semantic-wise self-distillation to balance the optimization objective of each modality. Then, we define relative preference to evaluate the dominance of each modality during training, based on which to design task-wise and gradient-wise regularization to balance the convergence rates of different modalities. Experimental results on two publicly available multi-modal datasets demonstrate the superiority of PASSION against existing approaches for modality balancing. More importantly, PASSION is validated to work as a plug-and-play module for consistent performance improvement across different backbones. Code will be available upon acceptance.

## CCS CONCEPTS

• **Computing methodologies → Image segmentation**.

## KEYWORDS

Multi-modality image segmentation, Imbalanced missing rate, Cross-modality interaction, Prototype learning

## 1 INTRODUCTION

Multiple imaging modalities are widely used in clinical practice like multi-parametric magnetic resonance imaging (MRI) [2–4]. Such homogeneous multi-modal data representing the same media type provides various tissue contrast views and spatial resolutions, making it possible to quantitatively categorize histological sub-regions [9, 43]. Nevertheless, due to factors like image degradation, patient motion-related artifacts, incorrect acquisition settings, and cost constraints, MRI sequences may be incomplete [15, 17, 22]. Consequently, despite the great success of multi-modal learning on medical segmentation [7, 16, 20, 41, 43], it can hardly be directly deployed in practice.

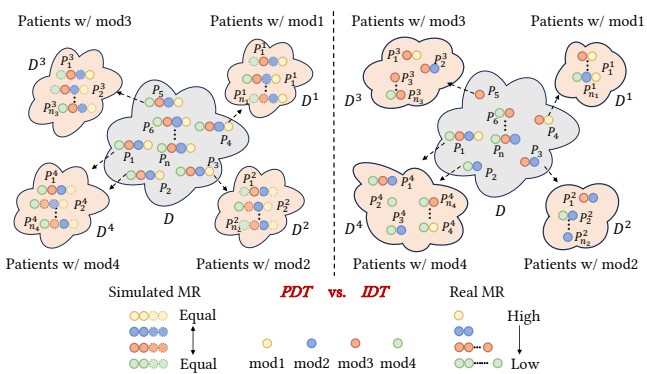

**Figure 1: Comparison of modality settings in incomplete multi-modal medical image segmentation.** *PDT* **denotes perfect data training where modalities share equal missing rates and masked modalities can be visible during training.** *IDT* **denotes imperfect data training, formulated in this paper, where modalities own imbalanced missing rates and missing modalities are invisible during training. Incomplete multi-modal sets** $D^1, D^2, D^3$, **and** $D^4$ **are generated/simulated from full-modality data** $D$ **in** *PDT* **and drawn from incomplete-modality data** $D$ **following imbalanced missing rates in** *IDT*.

To deal with missing modalities, various approaches have been proposed for incomplete multi-modal segmentation. One straightforward strategy is to synthesize missing modalities with available full-modality data through generative adversarial networks [14], which achieves full-modal segmentation with synthetic images [25, 32, 42]. Another typical solution is to transfer knowledge from full-modality networks to missing-modality ones [1, 39], which requires superior teacher models trained with full-modality data and dedicates student models for each combination of missing modalities. Alternatively, another efficient approach is shared representation learning, which aims to train a unified segmentation model by sharing a spatial mapping across modalities with multiple modality-specific encoders and a shared decoder [5, 9, 10, 33, 44]. Compared to the former two methodologies, it provides greater flexibility with lower computational complexity and thus becomes the state-of-the-art (SOTA) paradigm.

Unfortunately, previous studies tend to focus on deployment efficiency while idealizing the training process. In other words, they all assume perfect data training (*PDT*) where data is available with full-modality. As illustrated in Figure 1, in *PDT*, incomplete multi-modal data is generated by randomly masking modalities with equal probabilities. One typical setting is to randomly mask modalities during each training round, making invisible modalities in one round visible in another round. The other setting is to randomly mask modalities before training and keep masked modalities invisible. Both settings assume different modalities share the same

missing rates. Nevertheless, imperfect data exists not only in deployment but also during data collection and model training [21], which is under-explored. Furthermore, the missing rates of different modalities may vary in practical scenarios [34]. For instance, in MRI modalities, T1 is the most commonly acquired while T2 usually is unavailable due to its sensitivity to artifacts [15], resulting in situations where modalities are not equally available for model training and clinical evaluation [6, 19]. Inspired by this, we, for the first time, formulate such a scenario as **imperfect data training (**$IDT$**) where modalities own imbalanced missing rates and missing modalities are kept invisible throughout training**, as illustrated in Figure 1. Such imbalanced missing rates bring modality imbalance, where weak modalities (*i.e.*, with higher missing rates) may be dominated by strong modalities (*i.e.*, with lower missing rates), suffering from being under-trained. Despite great efforts on modality re-balancing [12, 28], how to balance unequal training in $IDT$ for multi-modal segmentation has not been exploited.

Under modality absence in model training, existing approaches encounter severe issues. Relying on full-modality training data as ground truth makes it challenging to supervise the reconstruction task in image synthesis [32, 42] and to train teacher models for knowledge transfer [1, 39]. In shared representation learning, the unified multi-modal model typically is trained by masking modality-specific features in the fusion stage. However, during training, such "masked" modality data still plays a role in regularizing the uni-modal performance [9, 33, 44]. Consequently, it would suffer from imbalanced regularization for weak modalities in $IDT$.

Based on the observation that, in multi-modal segmentation, multi-modal always performs better than uni-modal, and the knowledge gap between multi-modal and each uni-modal somewhat reflects modality dominance, we propose Preference-Aware Self diStillatION (PASSION) transferring multi-modal knowledge to all available uni-modals under the guidance of modality preference. Specifically, pixel-wise and semantic-wise self-distillation is first constructed to regularize the optimization objective of each uni-modal in a unified framework. Compared to training via ground truth, it would re-balance each modality according to its distance to multi-modal knowledge. Then, considering the varying learning/optimization paces of different modalities, we define relative preference to estimate how strong each uni-modal is compared to others. Then, task-wise and gradient-wise preference-aware regularization is constructed to ensure a balanced convergence of different modalities. The main contributions are as follows:

- We identify and formulate a more realistic and challenging task — incomplete multi-modality medical image segmentation with imbalanced missing rates. To our knowledge, it is the first exploration in multi-modal image segmentation.
- We propose a new self-distillation solution PASSION to balance different modalities in both optimization objectives and paces. In addition, PASSION is proven, through extensive evaluation, to work as a plug-and-play module for performance improvement across various backbones.
- We prove the superiority of PASSION against SOTA modality-balancing approaches through comprehensive experiments on BraTS2020 and MyoPS2020 for incomplete multi-modal segmentation with imbalanced missing rates.

## 2 RELATED WORK

### 2.1 Incomplete Multi-modal Segmentation

Incomplete data is a common and long-standing issue in practical scenarios, particularly in clinical practice. The most commonly studied task is incomplete multi-modal medical image segmentation usually involving MRI images with various missing components [17]. As standard multi-modal segmentation methods built on full-modality data [20, 43] would encounter severe performance degradation given only incomplete modalities, it is of great value but also challenging.

To pursue effective multi-modal medical image segmentation, extensive research has been broadly explored including image synthesis, knowledge transfer, and shared representation learning. Specifically, Sharma and Hamarneh [32] used a U-Net [8, 31] generator to impute missing modalities, with a generative adversarial network [14] learning to discriminate between real and synthesized inputs. Ma et al. [25] proposed a meta-learning algorithm [13] to reconstruct the features of missing modalities. Wang et al. [39] trained a teacher-student framework for each subset of modalities. Azad et al. [1] proposed a style-matching mechanism to reconstruct missing information from a full-modality network. Chen et al. [6], Wang et al. [36] trained uni-modal models by transferring privileged knowledge from a full-modal teacher to each uni-modal student. Liu et al. [23] focused on self-supervised pre-training. However, in $IDT$, there may not exist sufficient full-modality data as a target for image synthesis and full-modal teacher training, making them hard to deploy in real-world training.

To overcome such limitations, Havaei et al. [17] and Dorent et al. [10] computed variational statistics to construct a uniform representation for segmentation. Chen et al. [5] and Ding et al. [9] performed shared representation fusion by modality re-weighting and attention-gating modules. Zhao et al. [45] introduced graph convolutional networks for incomplete brain tumor segmentation. Wang et al. [35] learned shared and specific features by distribution alignment and domain classification. Zhang et al. [44] and Shi et al. [33] introduced transformers to exploit both intra- and inter-modal dependence for feature fusion. Qiu et al. [30] proposed a group self-support algorithm to make use of the sensitive modality's specific features. However, such shared representation learning-based methods were proposed only for incomplete-modality inference while ignoring data missing in training. In other words, the frequency of each modality during training remains the same. Konwer et al. [21] discussed the limited full-modality data scenario and proposed a meta-learning method to enhance modality representations. Unfortunately, how to pursue effective multi-modal medical image segmentation with imbalanced missing rates is under-explored, which is more valuable in clinical practice.

### 2.2 Imbalanced Multi-modal Learning

Due to modality discrepancy, multi-modal learning naturally encounters the concern of fairness and imbalance. Wang et al. [37] found that different modalities could overfit and generalize at different rates and thus obtain sub-optimal solutions when jointly training them using a unified optimization strategy. Peng et al. [28] proposed that the better-performing modality would dominate gradient updating while suppressing the learning process of other

modalities. To address such issues, several approaches have been proposed to deal with modal imbalance. Du et al. [11] proposed to improve uni-modal performance through knowledge distillation from well-trained models. Xiao et al. [40] proposed to randomly drop the audio pathway during training as a regularization technique to adjust the learning paces between visual and audio pathways. Wang and Singh [38] proposed a re-weighting scheme by assigning lower weights to data samples with missing values to promote fairness. Peng et al. [28] chose to slow down the learning rate of the mighty modality by online modulation to lessen the inhibitory effect on other modalities. Fan et al. [12] performed stimulation on the particularly slow-learning modality by enhancing its clustering towards prototypes. Sun et al. [34] proposed the concept of relative advantage for modalities and adaptively regulated the supervision of all modalities during training. Though existing works all focus on classification instead of segmentation, they motivate us to pursue effective incomplete multi-modal medical image segmentation via re-balancing the convergence rates of different modalities in $IDT$.

## 3 METHODOLOGY

### 3.1 Problem Definition

Given a multi-modal training dataset containing $N$ samples, each sample comprises $M$ modalities. Let $[P]$ for any positive integer $P$ denote the set $\{1, 2, \ldots, P\}$. Each training sample is denoted as $(\{x_n^m\}_{m \in [M]}, \{y_n^m\}_{m \in [M]})$, where $n \in [N]$ is sample index. For a segmentation task, it is to assign each pixel with an individual category label from $K$ categories. To formulate this, we use $(x_{n,i}^m, y_{n,i}^m)$ to denote the raw input and label of pixel $i$ corresponding to modality $m$ and sample $n$. For multi-modal segmentation studied in this paper where all modalities share a common label, $y_{n,i} = y_{n,i}^1 = y_{n,i}^2 = \cdots = y_{n,i}^M$ holds. Let $C \in \mathbb{R}^{N \times M}$ be the modality presence indication matrix, and let $C_{nm}$ denote the $(n, m)$-th entry of $C$. Then $C_{nm}$ is 1, if modality $m$ of sample $n$ is available; otherwise $C_{nm}$ equals to 0. The missing rate $MR$ of modality $m$, $\forall m \in [M]$, is calculated as $MR^m = (N - \sum_{n \in [N]} C_{nm})/N$. We assume $MR^m \in [0, 1)$ to ensure that at least one sample of each modality is available during training.

**Incomplete Multi-modal Segmentation Baseline.** In existing research, the SOTA paradigm [9, 33, 44] adopts $m$ modality-specific encoders $E_m$ and a shared fusion decoder $D_f$. Layer-wise features extracted from available $E_m$ are skipped like U-Net to be fused in $D_f$. For the consistency of expression, we use $z_n^l = D_f^l(x_n)$ as the fused features of all available modalities of sample $n$ in layer $l$, and $z_n^{m,l} = D_f^l(x_n^m)$ as the features of only modality $m$. Denoting $l = 0$ as the output layer, the overall objective of the paradigm is

$$\mathcal{L}_{task} = \sum_{l=0}^{L} \ell_{dice+ce}(\text{Up}_{2^l}(z_n^l), y_n) + \sum_{m \in [M]} \ell_{dice+ce}(z_n^{m,0}, y_n), \quad (1)$$

where $\ell_{dice+ce}$ denotes the Dice and weighted cross-entropy loss and $\text{Up}_{2^l}$ represents $2^l \times$ upsampling. For simplification, we omit the low-dim transform and softmax operation $\sigma$. In $\mathcal{L}_{task}$, the former item imposes the final prediction objective and deep supervision, and the latter forces each modality to attain its optimal task-relevant representations. Denoting them $\mathcal{L}_{seg}$ and $\mathcal{L}_{reg}$ respectively, the

overall objective in $IDT$ is re-formulated as:

$$\mathcal{L}_{task} = \mathcal{L}_{seg} + \sum_{\substack{m \in [M] \\ C_{nm}=1}} \mathcal{L}_{reg}^m, \quad (2)$$

where $\mathcal{L}_{reg}^m$ denotes the regularization term of modality $m$ in $\mathcal{L}_{reg}$. Such an objective begs a problem: We cannot optimize unavailable modalities in $IDT$. In other words, it will result in a severe imbalance across modalities given imbalanced missing rates, as $\mathcal{L}_{reg}$ **would emphasize more on modalities with lower missing rates for loss minimization**. What's worse, as each single modality is not well optimized, it may cause severe fusion issues in the SOTA paradigm. Such observations motivate us to balance each modality's performance during unified end-to-end training given imbalanced modality missing rates.

### 3.2 Multi-Uni Self-Distillation

Motivated by knowledge distillation (KD) [18], which aims to transfer the teacher's "dark knowledge" to students via soft labels, we treat multi-modal knowledge as a common objective for each available uni-modal to balance inter-modal learning. As multi-modal knowledge is learned through all modalities, it may be dominated by certain modalities of lower missing rates. **When penalized through KD, such modalities would be less emphasized as they are closer to multi-modal knowledge (*i.e.*, soft labels)**. In this way, it is promising to re-balance modalities in $IDT$. Unlike previous KD-based works relying on a separate complete multi-modal teacher based on $PDT$ [1, 6, 36, 39], we prefer a unified network to transfer knowledge from multi-modal to uni-modal, denoted as multi-uni self-distillation. On the one hand, it reduces the difficulty of training a sufficiently robust teacher model under $IDT$. On the other hand, multi-modal knowledge is innate but under-exploited for uni-modal in a unified framework. Specifically, multi-uni self-distillation is composed of pixel-wise and semantic-wise self-distillation described in the following.

**Pixel-wise Self-Distillation.** As segmentation can be formulated as a pixel-level classification task, we propose to align the predictions of each pixel between multi-modal and uni-modal. It is based on the observation, validated through experiments, that feature-level alignment usually results in multi-modal performance degradation while logit alignment in deep supervision is a more robust option for imbalanced learning [26]. Therefore, pixel-wise multi-uni self-distillation for each uni-modal $m$ is formulated as:

$$\mathcal{L}_{pixel}^m = \sum_{l=0}^{L} KL[\sigma(\frac{z_n^{m,l}}{\tau}) || \sigma(\frac{z_n^l}{\tau})], \quad (3)$$

where $KL$ denotes the Kullback-Leibler divergence, $\tau$ is the temperature hyper-parameter, $\sigma$ is the softmax function, and $z_n^l$ and $z_n^{m,l}$ represent features of multi-model and modality $m$ respectively from layer $l$ of sample $n$.

**Semantic-wise Self-Distillation.** By learning more information from diverse modalities, the multi-modal teacher captures more robust intra- and inter-class representations [36]. Therefore, not only local pixel-wise knowledge but also global class-wise knowledge should be transferred to uni-modal. By using prototypes to represent the general features of a class, we hope to build multi-modal and uni-modal prototypes to achieve global knowledge transfer.

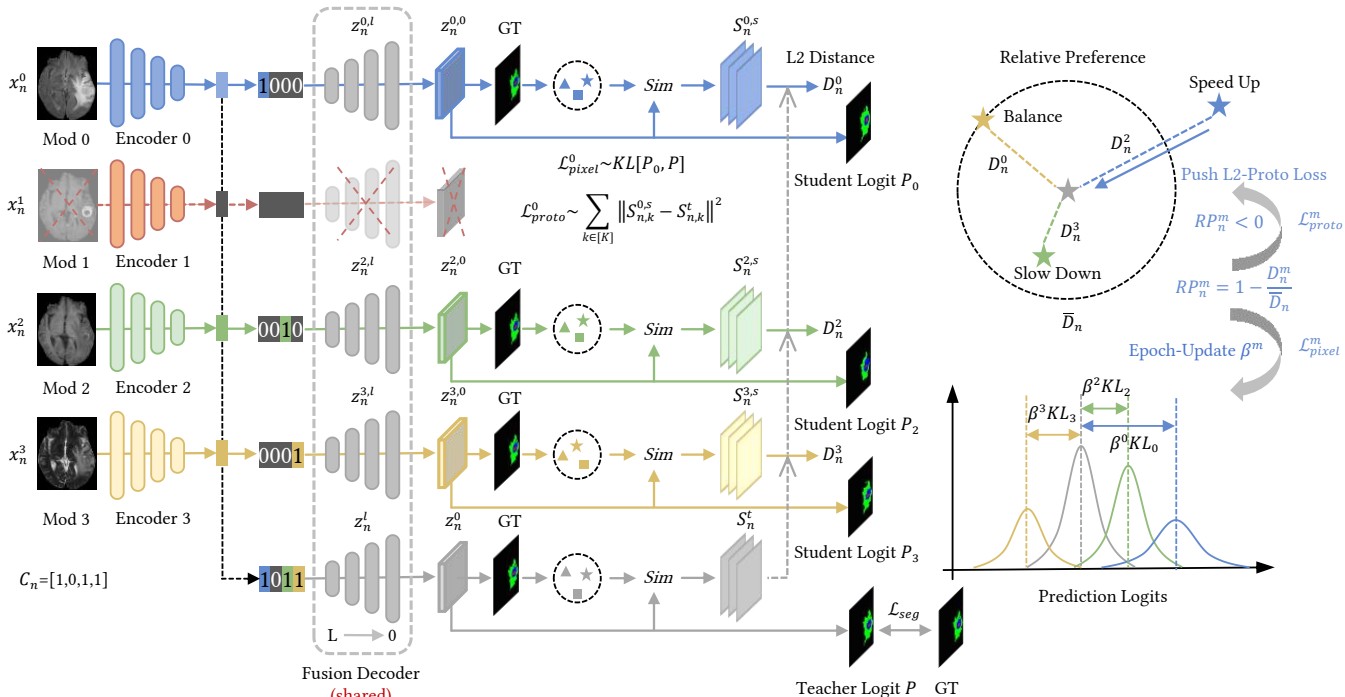

**Figure 2: Illustration of PASSION. $\mathcal{L}_{proto}^m$ and $\mathcal{L}_{pixel}^m$ represent multi-uni self-distillation and $\delta^m$ and $\beta^m$ represent preference-aware regularization. ▲, ★, and ■ represent prototypes in Eq. 4 while $Sim$ denotes feature-prototype similarity in Eq. 5**

Here, the prototype of each sample is calculated individually, as each sample contains sufficient pixels in segmentation.

The prototypes of class $k$ for sample $n$ corresponding to the teacher $c_{n,k}^t$ and any student $c_{n,k}^{m,s}$ are calculated by

$$
\begin{aligned}
c_{n,k}^t &= \frac{\sum_i z_{n,i}^0 \mathbb{1}[y_{n,i} = k]}{\sum_i \mathbb{1}[y_{n,i} = k]}, \\
c_{n,k}^{m,s} &= \frac{\sum_i z_{n,i}^{m,0} \mathbb{1}[y_{n,i} = k]}{\sum_i \mathbb{1}[y_{n,i} = k]},
\end{aligned}
\tag{4}
$$

where $z_{n,i}^0$ and $z_{n,i}^{m,0}$ represent the multi- and uni-modal (*i.e.*, modality $m$) output (*i.e.*, layer $l = 0$) features of pixel $i$ for sample $n$. $\mathbb{1}$ represents an indicator being 1 if the argument is true or 0 otherwise. To measure the semantic richness within $c_{n,k}^{m,s}$, we define $S_{n,k}^{m,s}$ to calculate the feature-prototype similarity for class $k$, defined as

$$
S_{n,k}^{m,s} = \sum_i Cos(z_{n,i}^{m,0}, c_{n,k}^{m,s}),
\tag{5}
$$

where $Cos(\cdot, \cdot)$ denotes cosine similarity. It should be noted that $S_{n,k}^{m,s}$ measures both intra- or inter-class semantic correlations depending on whether pixel $i$ belongs to class $k$ or not. Following this, the semantic richness of multi-modal prototypes $c_{n,k}^t$ for class $k$ is defined as

$$
S_{n,k}^t = \sum_i Cos(z_{n,i}^0, c_{n,k}^t).
\tag{6}
$$

Then, the semantic-wise knowledge gap across all classes between uni-modal $m$ and multi-modal is defined as

$$
D_n^m = \sum_i \sum_{k \in [K]} \left\| S_{n,k}^{m,s}(i) - S_{n,k}^t(i) \right\|_2,
\tag{7}
$$

where $\|\cdot\|_2$ denotes the $\ell_2$-norm. Accordingly, semantic-/class-wise knowledge transfer for modality $m$ is accomplished by minimizing

$$
\mathcal{L}_{proto}^m = \sum_i \sum_{k \in [K]} \left\| S_{n,k}^{m,s}(i) - S_{n,k}^t(i) \right\|_2^2.
\tag{8}
$$

Here, using an L2-like loss is to further up-weight weak modalities for balanced optimization.

### 3.3 Preference-Aware Regularization

As discussed above, multi-uni self-distillation equally transfers knowledge to available uni-modal to balance inter-modal learning. However, it may still make those modalities with higher missing rates struggle to keep up with others. This is because the recurring uni-modal wins at optimization more often than others. Therefore, dynamically evaluating how strong (or weak) each uni-modal is compared to others and balancing the learning paces across them in *IDT* is crucial.

**Relative preference.** Since different uni-modal students are born with unequal talents and expertise, measuring the learning progress of different students directly through their performance seems to be inappropriate. Noting that the multi-modal teacher usually prefers strong modalities that are easy to learn or have lower missing rates, we use the distance $D_n^m$ to represent the relative

preference of the multi-modal teacher to uni-modal $m$. It is based on the observation that the higher $D_n^m$ is, the more modality $m$ is neglected. Given any sample $n$, the average distance is calculated as:

$$\bar{D}_n = \sum_{\substack{m \in [M] \\ C_{nm}=1}} D_n^m / \sum_{m \in [M]} C_{nm}, \tag{9}$$

where $C_{nm}$ indicates the presence of modality $m$ in sample $n$ according to the modality presence indication matrix $C$ defined in Sec. 3.1. With $\bar{D}_n$, the relative preference to modality $m$ is formally defined as:

$$RP_n^m = 1 - \frac{D_n^m}{\bar{D}_n}. \tag{10}$$

When $RP_n^m > 0$ (resp., $< 0$), modality $m$ is more preferred (resp., neglected) compared to other modalities for sample $n$. Especially, $RP_n^m = 0$, if modality $m$ is at the balancing point or unavailable.

**Re-balancing regularization.** Given the relative preference $RP^m$ of modality $m$ for each sample, it is expected to slow down the learning paces of the preferred modalities and speed up the neglected ones. Inspired by previous works on imbalanced multi-modal learning [12, 28, 34], we develop two regularization items for each modality: task-wise and gradient-wise.

Noting that strong modalities occupy the center of gravity in early training due to their larger data amounts and easy learning properties. In task-wise, we propose to identify and push the slow-learning modalities based on sample-level relative preference. If $RP_n^m$ is negative, we should further accelerate the slow-learning modality $m$. Therefore, we set a task mask $\delta_n^m$. If modality $m$ is neglected, it should be accelerated for training by enhancing its semantic learning from multi-modal. Such a rule is formulated as:

$$\delta_n^m = \mathbb{1}\left[RP_n^m < 0\right], \tag{11}$$

where $\mathbb{1}$ represents an indicator being 1 if the argument is true or 0 otherwise. It should be noted that $\delta_n^m$ is to regularize semantic-wise self-distillation through $\mathcal{L}_{proto}^m$.

In terms of gradient-wise regularization, we propose to continually balance pixel-wise self-distillation through epoch-level relative preference. Let $\beta^m$ denote the gradient-weighted coefficient of modality $m$ which is initialized as $\frac{1}{1-MR^m}$ to balance the inequality in modality amounts. Then we calculate epoch-average relative preference

$$\bar{RP}^m = \frac{\sum_{n \in [N]} RP_n^m}{\sum_{n \in [N]} C_{nm}}, \tag{12}$$

and apply gradient descent to update $\beta^m$ every epoch by

$$\beta_{r+1}^m = \beta_r^m - \gamma \cdot \bar{RP}^m, \tag{13}$$

where $\gamma$ is the updating rate and $r$ indexes the epoch. For modality $m$ with $\bar{RP}^m > 0$ or $\bar{RP}^m < 0$, its learning speed will be decreased or increased accordingly.

## 3.4 Overall Objective

Combining multi-uni self-distillation and preference-aware regularization, the overall optimization objective is written as

$$\mathcal{L} = \mathcal{L}_{seg} + \sum_{\substack{m \in [M] \\ C_{nm}=1}} (\lambda_1 \beta^m \mathcal{L}_{pixel}^m + \lambda_2 \delta_n^m \mathcal{L}_{proto}^m), \tag{14}$$

where $\lambda_1$ and $\lambda_2$ are balancing hyper-parameters.

## 4 EXPERIMENTS

### 4.1 Datasets and Evaluation Metrics

Two segmentation tasks with publicly-available multi-sequence MRI datasets are adopted for evaluation, including:

(1) **BraTS2020** [27]: The multimodal brain tumor segmentation challenge (BraTS2020) dataset consists of 369 cases with ground truth labels and four MRI modalities (*i.e.*, T1, T1c, Flair, and T2). Ground truth is provided with the normal tissue and three tumor sub-regions including the necrotic and non-enhancing tumor core (NCR/NET), the peritumoral edema (ED), and GD-enhancing tumor (ET). Following the task setting in the challenge, tumor classes are merged into the whole tumor (WT) including all tumor sub-regions, tumor core (TC) consisting of NCR/NET and ET, and enhancing tumor involves ET. The dataset is split into 219, 50, and 100 cases for training, validation, and testing respectively.

(2) **MyoPS2020** [29, 46]: The MyoPS 2020 challenge dataset consists of 25 cases of multi-sequence CMR (i.e., bSSFP, LGE, and T2), with labels being provided for myocardial pathology segmentation. Targets include the normal tissue, left ventricular blood pool (LVB), right ventricular blood pool (RVB), normal myocardium, myocardial edema, and myocardial scars of the left ventricle. Following standard practice, the last three classes are grouped as myocardium of the left ventricle (MYO). The dataset is split into 20 and 5 cases with multi-slices for training and test respectively.

For image pre-processing, each brain MRI volume is re-sampled to the $1mm^3$ resolution and each cardiac CMR scan is resampled into the same spatial resolution. Following [6, 9], we cut out the black background areas outside the brain, center-crop the heart regions, and further normalize the intensity of each volume to zero mean and unit variance. Considering the specificity of two datasets, we perform the former task (*i.e.*, **BraTS2020**) in 3D and the latter (*i.e.*, **MyoPS2020**) in 2D to verify the flexibility of PASSION.

For evaluation, two most commonly-used metrics are selected, including the Dice similarity coefficient (*i.e.*, Dice) and Hausdorff distance (*i.e.*, HD). Dice measures the voxel-wise accuracy and HD evaluates the surface distance. Higher Dice and lower HD indicate better segmentation performance. Quantitative evaluation is performed on subject-level volume segmentation to be consistent with the BraTS and MyoPS challenges.

### 4.2 Implementation Details

For comparison against SOTA imbalanced multi-modal learning approaches, mmFormer [44] is set as the backbone while for plug-and-play evaluation another two SOTA multi-modal segmentation methods RFNet [9] and M2FTrans [33] are included as backbones. All models are implemented and modified with the same basic-dimension size in Pytorch and trained using the optimizer AdamW [24] with an initial learning rate of 2e-4, a weight decay of 1e-4, and a batch size of 1 on NVIDIA Geforce RTX 3090 GPUs for 300 epochs. Specifically, we adopt a poly decay strategy with $p = 0.9$ during training and set the temperature $\tau$ for pixel-wise self-distillation as 4 and the hyper-parameters $\lambda_1$ and $\lambda_2$ as 0.5 and 0.1 respectively for the self-distillation loss. The parameter setting for re-balancing

**Table 1: Quantitative comparison on BraTS2020 and MyoPS2020 under various settings.** *MR* is short for the missing rates of modalities (T1/T1c, Flair, T2) for BraTS2020 and (bSSFP, LGE, T2) for MyoPS2020 respectively. *s*, *m*, and *l* correspond to small, medium, and large, set as *s* = 0.2, *m* = 0.5, and *l* = 0.8 for BraTS2020 and *s* = 0.3, *m* = 0.5, and *l* = 0.7 for MyoPS2020 respectively. *PDT* denotes balanced modality distributions *MR* = (0, 0, 0) for perfect data training. Here, T1 and T1c share the same missing rate but are treated as separate modalities.

| MR | Dataset | BraTS2020 | | | | | | | | MyoPS2020 | | | | | | | |
|---|---|---|---|---|---|---|---|---|---|---|---|---|---|---|---|---|---|
| | Metric | DSC [%] ↑ | | | | HD [mm] ↓ | | | | DSC [%] ↑ | | | | HD [mm] ↓ | | | |
| | Method | WT | TC | ET | Avg. | WT | TC | ET | Avg. | LVB | RVB | MYO | Avg. | LVB | RVB | MYO | Avg. |
| PDT | Baseline | 83.66 | 73.18 | 54.91 | 70.58 | 14.99 | 17.18 | 11.88 | 14.68 | 84.61 | 58.08 | 80.13 | 74.27 | 5.90 | 15.46 | 7.63 | 9.66 |
| (s,m,l) | Baseline | 81.89 | 69.36 | 51.72 | 67.66 | 16.51 | 16.41 | 10.14 | 14.35 | 77.69 | 56.94 | 71.81 | 68.81 | 19.38 | 22.62 | 21.32 | 21.11 |
| | +ModDrop | 81.31 | 68.75 | 51.53 | 67.20 | 19.24 | 17.94 | 10.91 | 16.03 | 80.63 | 51.42 | 75.70 | 69.25 | 15.42 | 31.84 | 20.13 | 22.46 |
| | +PMR | 82.59 | 70.44 | **52.86** | 68.63 | 16.84 | 18.22 | 11.68 | 15.58 | 78.05 | 57.61 | 74.32 | 69.99 | 18.16 | 23.04 | 17.42 | 19.54 |
| | +**PASSION** | **83.42** | **71.74** | 52.78 | **69.31** | **11.98** | **11.78** | **7.80** | **10.52** | **81.44** | **60.97** | **77.44** | **73.28** | **11.36** | **20.49** | **11.64** | **14.50** |
| (s,l,m) | Baseline | 81.53 | 67.74 | 50.02 | 66.43 | 19.88 | 19.71 | 11.70 | 17.10 | 74.25 | 54.82 | 66.79 | 65.29 | 25.18 | 32.92 | 25.86 | 27.99 |
| | +ModDrop | 80.54 | 69.10 | 51.64 | 67.09 | 26.98 | 25.12 | 16.05 | 22.72 | 78.57 | 54.93 | 74.83 | 69.44 | 15.91 | 25.52 | 17.12 | 19.52 |
| | +PMR | 82.32 | 68.39 | 50.88 | 67.20 | 15.39 | 17.63 | 10.76 | 14.59 | 77.67 | 57.38 | 73.98 | 69.68 | 16.84 | 23.12 | 18.68 | 19.55 |
| | +**PASSION** | **83.47** | **71.57** | **53.06** | **69.37** | **10.52** | **11.21** | **7.20** | **9.64** | **80.20** | **58.22** | **77.56** | **71.99** | **16.23** | **18.13** | **13.01** | **15.79** |
| (m,s,l) | Baseline | 81.56 | 70.22 | 52.06 | 67.95 | 21.00 | 22.98 | 14.33 | 19.44 | 77.16 | 54.44 | 72.08 | 67.89 | 15.78 | 21.13 | 24.56 | 20.49 |
| | +ModDrop | 80.40 | 68.74 | 50.58 | 66.57 | 27.80 | 29.57 | 20.67 | 26.01 | 79.07 | 55.30 | **75.69** | 70.02 | 12.46 | 22.32 | 17.27 | 17.35 |
| | +PMR | 81.73 | 70.29 | 51.76 | 67.93 | 18.62 | 20.27 | 13.75 | 17.55 | 77.49 | 57.62 | 71.90 | 69.00 | 19.21 | 23.03 | 20.39 | 20.88 |
| | +**PASSION** | **83.09** | **70.80** | **52.82** | **68.90** | **10.47** | **12.30** | **8.35** | **10.37** | **80.55** | **64.56** | 74.53 | **73.21** | **10.74** | **18.69** | **14.95** | **14.79** |
| (m,l,s) | Baseline | 81.40 | 68.49 | 51.81 | 67.23 | 17.34 | 20.37 | 12.87 | 16.86 | 75.60 | 47.37 | 70.85 | 64.61 | 22.04 | 30.43 | 19.48 | 23.98 |
| | +ModDrop | 80.83 | 68.92 | 50.94 | 66.90 | 24.70 | 26.19 | 18.84 | 23.24 | 76.86 | 50.29 | 74.13 | 67.09 | 21.63 | 25.86 | 17.75 | 21.75 |
| | +PMR | 82.18 | 69.27 | 52.73 | 68.06 | 16.57 | 20.09 | 12.90 | 16.52 | 78.59 | 50.64 | 71.16 | 66.80 | 20.85 | 22.77 | 23.60 | 22.41 |
| | +**PASSION** | **83.41** | **70.49** | **53.27** | **69.06** | **12.12** | **14.42** | **10.10** | **12.21** | **79.80** | **54.06** | **75.86** | **69.91** | **15.59** | **21.95** | **15.42** | **17.65** |
| (l,s,m) | Baseline | 80.77 | 68.15 | 50.85 | 66.59 | 17.68 | 17.84 | 10.96 | 15.49 | 77.74 | 52.86 | 70.97 | 67.19 | 18.72 | 28.28 | 18.91 | 21.97 |
| | +ModDrop | 80.22 | 68.50 | 50.69 | 66.47 | 25.93 | 25.47 | 17.78 | 23.06 | 80.99 | 53.16 | 75.90 | 70.02 | 15.70 | **22.17** | 17.34 | 18.40 |
| | +PMR | 80.94 | 68.18 | 51.54 | 66.89 | 14.97 | 16.29 | 10.94 | 14.07 | 79.72 | 54.21 | 72.72 | 68.88 | 16.57 | 24.96 | 23.02 | 21.52 |
| | +**PASSION** | **82.69** | **69.88** | **52.35** | **68.31** | **11.27** | **13.99** | **9.05** | **11.44** | **81.91** | **55.04** | **77.52** | **71.49** | **13.25** | 22.63 | **14.71** | **16.86** |
| (l,m,s) | Baseline | 80.33 | 68.06 | 52.01 | 66.80 | 17.43 | 18.87 | 13.57 | 16.62 | 80.44 | 51.70 | 75.14 | 69.09 | 13.06 | 21.65 | 15.38 | 16.70 |
| | +ModDrop | 80.60 | 67.75 | 50.63 | 66.33 | 23.59 | 24.24 | 17.12 | 21.65 | 80.59 | 51.95 | 76.41 | 69.65 | 12.81 | 20.95 | 13.07 | 15.61 |
| | +PMR | 81.13 | 67.96 | 50.99 | 66.69 | 17.70 | 19.61 | 13.05 | 16.79 | 80.15 | 54.54 | 76.39 | 70.36 | 12.64 | 22.02 | 12.56 | 15.74 |
| | +**PASSION** | **82.55** | **69.11** | **52.06** | **67.91** | **11.88** | **14.43** | **8.81** | **11.71** | **81.25** | **57.65** | **76.67** | **71.86** | **11.92** | **20.66** | **12.34** | **14.97** |

supervision is γ = 0.01. During training, each brain volume is randomly cropped to 80×80×80 pixels, each cardiac scan is center-cropped to 256×256 pixels, and both are then augmented by random rotation and intensity change. To emulate modality-imperfect data training, we randomly mask each sample's modalities according to the corresponding missing rates before training, directly drop those masked-modality inputs (*i.e.*, invisible), and set corresponding encoded features as all-zero vectors in training.

## 4.3 Evaluation on BraTS2020

**Quantitative evaluation.** Quantitative comparison results by introducing SOTA modality-balancing approaches to the baseline (*i.e.*, mmFormer [44]) on BraTS2020 are summarized in Table 1. Compared to PDT, segmentation under imbalanced modality missing rates is more challenging, resulting in apparent performance degradation of the baseline. Through modality re-balancing, PMR [12] achieves marginal performance improvements under most settings while ModDrop [40] even brings negative effects. This is because modalities are imbalanced not only in data amounts but also in semantics. For instance, Flair and T2 are much more informative than T1/T1c. Given relatively limited Flair and T2, it is more challenging to explore sufficient semantic information, making both PMR and ModDrop fail. Simply re-balancing modalities without considering semantic misalignment can be counter-productive.

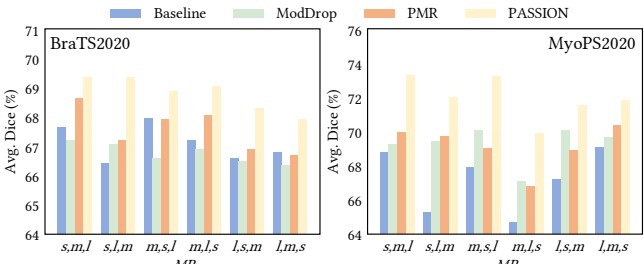

**Figure 3: Visualized average performance comparison on BraTS2020 and MyoPS2020 given (*s* = 0.2, *m* = 0.5, *l* = 0.8) and (*s* = 0.3, *m* = 0.5, *l* = 0.7) respectively.**

Comparatively, through multi-uni self-distillation to re-balance loss penalization and preference-aware re-balancing regularization to adaptively align modality learning speeds, PASSION effectively re-balances modalities for feature extraction, leading to consistent performance improvements under various modality missing rates. Visualized quantitative comparison is plotted in Figure 3, showing the superiority of PASSION for modality balancing in *IDT*.

**Qualitative evaluation.** Exemplar segmentation results under different modality combinations on BraTS2020 are illustrated in Figure 4. Given only T2 (*i.e.*, the fewest modality), all comparison

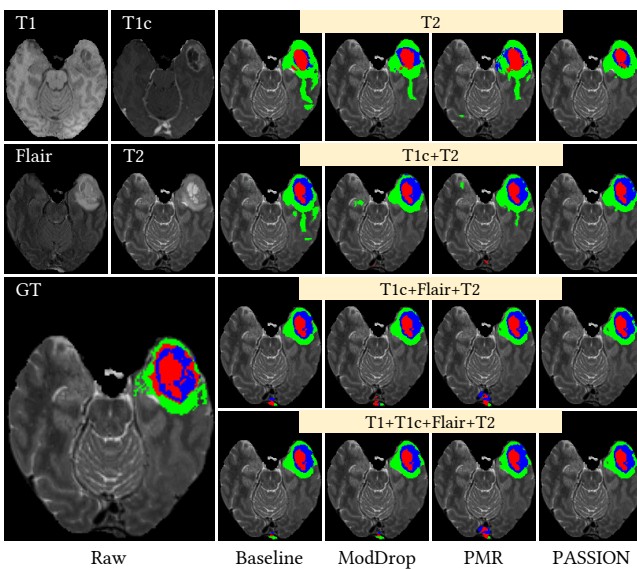

**Figure 4: Exemplar segmentation results on BraTS2020 under four modality combinations given** $MR = (0.2, 0.4, 0.6, 0.8)$ **for T1, T1c, Flair, and T2 respectively.**

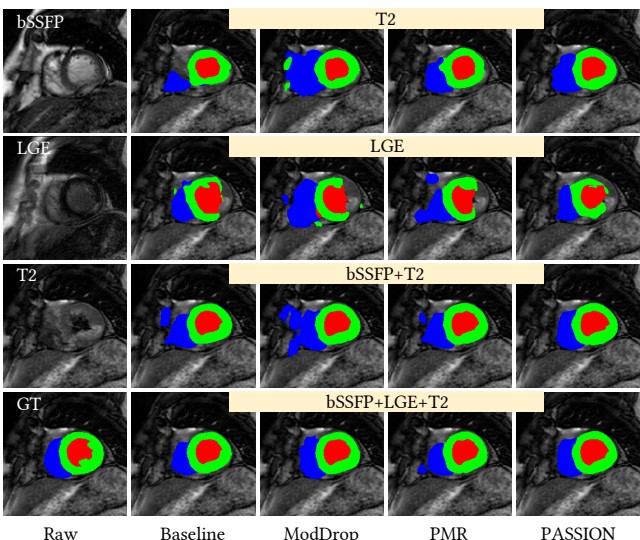

**Figure 5: Exemplar segmentation results on MyoPS2020 under three modality combinations given** $MR = (0.3, 0.5, 0.7)$ **for bSSFP, LGE, and T2 respectively.**

approaches produce extensive false positives. With more modalities introduced, segmentation performance is gradually improved, indicating imbalanced learning, especially for T2. Comparatively, PASSION significantly outperforms comparison approaches by effectively reducing false positives. More importantly, relatively stable/consistent performance across different modality combinations shows its effectiveness in modality re-balancing in $IDT$.

## 4.4 Evaluation on MyoPS2020

**Quantitative evaluation.** Quantitative comparison results on MyoPS2020 are summarized in Table 1. Different from BraTS2020, modality-wise semantic information across modalities is closer in MyoPS2020. In other words, modalities contribute more equally to different sub-types instead of certain modalities being dominant for specific sub-types. As a result, both ModDrop and PMR achieve noticeable performance improvements by modality re-balancing. Comparatively, PASSION consistently achieves greater performance improvements under all modality settings, demonstrating its superiority in exploring richer semantic information. Visualized quantitative comparison in Figure 3 further validates this.

**Qualitative evaluation.** Exemplar segmentation results on MyoPS2020 are illustrated in Figure 5. Compared to BraTS2020, segmentation performance across different settings is more consistent even given only T2 (*i.e.*, the least modality), indicating more equal semantic contributions of modalities. With more modalities, more false positives are recalled while more false positives are introduced. Comparatively, PASSION achieves the most consistent segmentation performance with the fewest false positives and false negatives.

## 4.5 Component-wise Ablation Study

We first validate the concept of relative preference defined in Section 3.3 by plotting $RP$ curves with and without PASSION and

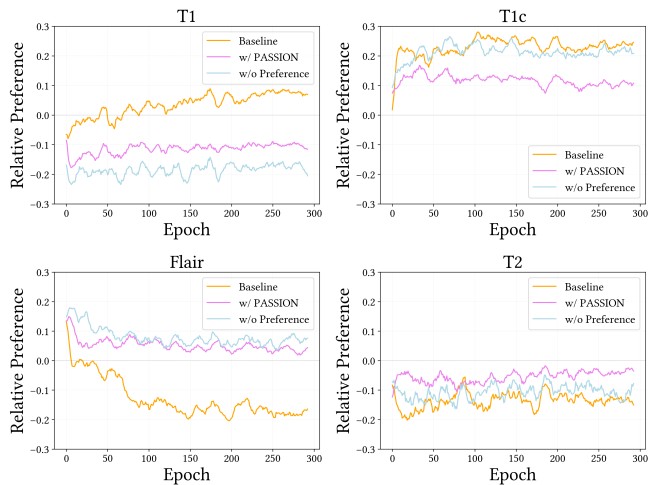

**Figure 6:** $RP$ **curves of baseline, baseline with PASSION, and baseline with modified PASSION by removing preference-aware regularization during** $IDT$ **training on BraTS2020 given** $MR = (0.2, 0.4, 0.6, 0.8)$**.**

preference-aware regularization during training as illustrated in Figure 6. It should be noted that the closer $RP$ approaches 0, the more balanced modalities are. Given only multi-uni self-distillation, modalities are marginally balanced, making $RP$ curves closer to 0. Comparatively, PASSION effectively re-balances different modalities with greater improvements.

Component-wise ablation study on BraTS2020 and MyoPS2020 is summarized in Table 2. In general, introducing each component separately is beneficial. Directly combining pixel-wise and semantic-wise self-distillation brings marginal performance improvements

**Table 2: Component-wise ablation study on BraTS2020 and MyoPS2020.**

| Component | | | | BraTS2020 $MR = (0.2, 0.4, 0.6, 0.8)$ | | | | | | | | MyoPS2020 $MR = (0.3, 0.5, 0.7)$ | | | | | | | |
|---|---|---|---|---|---|---|---|---|---|---|---|---|---|---|---|---|---|---|---|
| | | | | DSC [%] ↑ | | | | HD [mm] ↓ | | | | DSC [%] ↑ | | | | HD [mm] ↓ | | | |
| $\mathcal{L}_{pixel}$ | $\mathcal{L}_{proto}$ | $\delta$ | $\beta$ | WT | TC | ET | Avg. | WT | TC | ET | Avg. | LVB | RVB | MYO | Avg. | LVB | RVB | MYO | Avg. |
| ✗ | ✗ | ✗ | ✗ | 80.03 | 68.26 | 50.29 | 66.19 | 27.06 | 22.30 | 13.09 | 20.82 | 77.69 | 56.94 | 71.81 | 68.81 | 19.38 | 22.62 | 21.32 | 21.11 |
| ✓ | ✗ | ✗ | ✗ | 83.67 | 70.47 | 51.67 | 68.60 | 12.61 | 12.62 | 9.40 | 11.54 | 80.72 | 53.60 | 77.02 | 70.45 | 14.29 | 28.73 | 18.52 | 20.51 |
| ✗ | ✓ | ✗ | ✗ | 82.30 | 69.96 | 52.19 | 68.15 | 19.11 | 18.22 | 10.99 | 16.11 | 78.75 | 57.25 | 74.13 | 70.04 | 15.43 | 26.79 | 16.25 | 19.49 |
| ✓ | ✓ | ✗ | ✗ | 83.39 | 70.37 | 52.15 | 68.64 | 12.59 | 13.85 | 9.25 | 11.90 | 80.89 | 57.69 | 76.99 | 71.86 | 12.79 | 25.64 | 13.10 | 17.18 |
| ✓ | ✓ | ✓ | ✗ | 83.73 | 71.10 | 52.03 | 68.95 | 12.48 | 12.80 | 9.35 | 11.54 | 80.52 | 59.61 | 76.36 | 72.16 | 13.25 | 21.55 | 13.22 | 16.01 |
| ✓ | ✓ | ✓ | ✓ | **83.91** | **71.15** | **52.77** | **69.28** | **11.92** | **11.78** | **8.42** | **10.71** | **81.44** | **60.97** | **77.44** | **73.28** | **11.36** | **20.49** | **11.64** | **14.50** |

**Table 3: Quantitative comparison on various backbones evaluated on BraTS2020 under the *IDT* (*i.e., imperfect data training*) setting $MR = (0.2, 0.4, 0.6, 0.8)$ and the *PDT* (*i.e., perfect data training*) setting $MR = (0, 0, 0, 0)$ for T1, T1c, Flair, and T2 respectively.**

| Type | Setting | | T1 ● ○ ○ ○ ● ● ● ○ ○ ○ ● ● ● ○ ● | | | | | | | | | | | | | | | Avg. |
|---|---|---|---|---|---|---|---|---|---|---|---|---|---|---|---|---|---|---|---|
| | | T1c ○ ○ ● ○ ● ○ ○ ● ● ○ ● ● ○ ● ● | | | | | | | | | | | | | | | | | |
| | | Flair ○ ○ ● ○ ○ ● ○ ● ○ ● ● ○ ● ○ ● | | | | | | | | | | | | | | | | | |
| | | T2 ○ ○ ○ ● ○ ○ ● ○ ● ● ○ ● ● ● ● | | | | | | | | | | | | | | | | | |
| WT | PDT | RFNet | 68.53 | 69.27 | 82.28 | 82.25 | 74.07 | 85.51 | 84.90 | 85.18 | 83.94 | 86.42 | 86.61 | 85.40 | 87.63 | 87.36 | 88.30 | 82.51 |
| | | mmFormer | 69.60 | 69.87 | 83.34 | 83.33 | 74.74 | 86.60 | 85.81 | 87.39 | 85.47 | 87.73 | 87.99 | 86.40 | 88.60 | 88.84 | 89.27 | 83.67 |
| | | M2FTrans | 67.99 | 67.23 | 79.72 | 80.05 | 73.40 | 85.54 | 83.03 | 83.94 | 82.84 | 86.66 | 86.40 | 84.23 | 87.57 | 87.38 | 87.86 | 81.59 |
| | IDT | RFNet | 69.46 | 69.95 | 78.79 | 72.82 | 75.82 | 85.04 | 78.77 | 84.12 | 77.89 | 83.16 | 86.23 | 81.95 | 86.25 | 85.00 | 86.85 | 80.14 |
| | | mmFormer | 65.93 | 68.07 | 78.10 | 72.16 | 76.40 | 83.60 | 81.71 | 83.86 | 78.17 | 83.24 | 86.39 | 84.28 | 85.98 | 85.23 | 87.36 | 80.03 |
| | | M2FTrans | 52.02 | 69.24 | 80.26 | 71.39 | 76.27 | 85.92 | 81.14 | 84.26 | 78.91 | 81.68 | 86.71 | 83.15 | 86.19 | 84.91 | 87.29 | 79.29 |
| | IDT (+**PASSION**) | RFNet | 72.20 | 75.02 | 83.95 | 81.35 | 76.45 | 86.78 | 82.38 | 85.57 | 82.60 | 86.85 | 87.46 | 83.04 | 87.91 | 87.45 | 88.20 | **83.15** |
| | | mmFormer | 71.78 | 73.73 | 84.37 | 82.27 | 77.93 | 86.93 | 84.05 | 87.14 | 84.78 | 86.85 | 88.07 | 85.88 | 87.58 | 88.44 | 88.88 | **83.91** |
| | | M2FTrans | 71.95 | 74.19 | 84.26 | 80.84 | 77.59 | 86.79 | 82.62 | 86.31 | 83.85 | 85.65 | 87.18 | 83.99 | 86.59 | 86.59 | 87.06 | **83.03** |
| TC | PDT | RFNet | 59.53 | 77.24 | 64.30 | 66.77 | 81.45 | 70.28 | 70.39 | 80.16 | 81.90 | 70.70 | 81.67 | 83.28 | 72.83 | 81.97 | 82.80 | 75.02 |
| | | mmFormer | 56.00 | 76.40 | 61.74 | 63.74 | 80.50 | 67.07 | 66.61 | 79.67 | 81.36 | 68.66 | 80.71 | 82.23 | 69.89 | 81.25 | 81.90 | 73.18 |
| | | M2FTrans | 57.91 | 74.68 | 58.82 | 64.47 | 79.59 | 67.69 | 68.28 | 77.97 | 80.68 | 68.73 | 81.56 | 82.06 | 71.32 | 81.16 | 82.31 | 73.15 |
| | IDT | RFNet | 55.98 | 73.35 | 50.86 | 46.39 | 80.04 | 63.04 | 58.69 | 76.43 | 77.52 | 56.65 | 78.50 | 80.07 | 64.72 | 76.86 | 79.89 | 67.93 |
| | | mmFormer | 53.47 | 72.29 | 52.91 | 49.46 | 81.13 | 60.50 | 59.18 | 74.66 | 78.45 | 59.60 | 78.50 | 81.81 | 63.73 | 77.74 | 80.53 | 68.26 |
| | | M2FTrans | 35.94 | 73.44 | 52.38 | 39.83 | 80.99 | 62.54 | 57.99 | 76.83 | 78.30 | 52.32 | 80.22 | 81.64 | 63.97 | 77.03 | 80.92 | 66.29 |
| | IDT (+**PASSION**) | RFNet | 57.20 | 77.25 | 58.60 | 53.69 | 79.83 | 65.57 | 61.70 | 78.38 | 78.16 | 61.54 | 79.54 | 80.00 | 66.50 | 78.88 | 80.79 | **70.51** |
| | | mmFormer | 54.77 | 77.35 | 61.05 | 53.32 | 80.86 | 66.58 | 60.33 | 78.78 | 80.31 | 63.03 | 80.14 | 81.81 | 67.37 | 80.03 | 81.48 | **71.15** |
| | | M2FTrans | 57.03 | 77.39 | 58.63 | 52.83 | 80.21 | 65.33 | 62.21 | 78.03 | 79.40 | 62.49 | 79.58 | 80.49 | 67.50 | 78.98 | 80.34 | **70.70** |
| ET | PDT | RFNet | 31.99 | 65.81 | 36.67 | 40.08 | 68.56 | 41.19 | 43.61 | 68.38 | 69.18 | 43.62 | 69.47 | 71.04 | 45.21 | 68.76 | 69.64 | 55.55 |
| | | mmFormer | 28.66 | 67.25 | 33.76 | 38.20 | 70.70 | 38.51 | 41.02 | 68.11 | 69.83 | 42.17 | 70.86 | 70.82 | 43.22 | 69.91 | 70.62 | 54.91 |
| | | M2FTrans | 29.78 | 66.30 | 33.45 | 39.12 | 70.93 | 38.53 | 41.38 | 68.16 | 69.96 | 42.39 | 69.95 | 70.80 | 44.21 | 69.42 | 70.93 | 55.02 |
| | IDT | RFNet | 28.19 | 63.85 | 18.22 | 25.66 | 69.56 | 36.98 | 33.35 | 67.42 | 71.20 | 22.04 | 70.19 | 69.09 | 29.88 | 68.92 | 69.94 | 49.63 |
| | | mmFormer | 28.09 | 65.26 | 20.22 | 23.94 | 71.50 | 36.82 | 33.86 | 66.93 | 67.74 | 26.74 | 69.25 | 72.20 | 31.55 | 67.93 | 70.39 | 50.29 |
| | | M2FTrans | 26.63 | 65.43 | 13.37 | 21.00 | 71.89 | 34.45 | 33.20 | 66.28 | 67.53 | 24.96 | 70.38 | 71.92 | 37.62 | 67.91 | 70.62 | 49.55 |
| | IDT (+**PASSION**) | RFNet | 30.42 | 69.15 | 26.40 | 32.13 | 70.34 | 36.62 | 37.26 | 68.60 | 69.52 | 31.47 | 72.24 | 69.90 | 37.48 | 70.18 | 71.35 | **52.87** |
| | | mmFormer | 29.57 | 70.50 | 27.54 | 31.17 | 72.56 | 37.61 | 35.30 | 67.65 | 70.20 | 30.26 | 70.87 | 70.97 | 36.93 | 69.97 | 70.50 | **52.77** |
| | | M2FTrans | 29.93 | 67.53 | 28.40 | 32.18 | 70.86 | 37.01 | 37.28 | 67.18 | 69.15 | 36.55 | 68.22 | 70.14 | 39.62 | 67.68 | 69.82 | **52.77** |

without further regularization. Comparatively, jointly adopting all components leads to the best segmentation performance, greatly outperforming the baseline.

### 4.6 Plug-and-Play Ablation Study

PASSION is expected to work as a plug-and-play module onto various backbones for modality re-balancing. To validate this, PASSION is integrated into SOTA incomplete multi-modal medical image segmentation methods and evaluated under various modality missing rates, as summarized in Table 3. Compared to *PDT*, there exists noticeable performance degradation under *IDT* for all backbones. By introducing PASSION, consistent performance improvements across different sub-types under various modality combinations are achieved. One interesting observation is that adopting PASSION to

address *IDT* even outperforms *PDT* for WT segmentation, proving the robustness and flexibility of PASSION.

## 5 CONCLUSION

In this paper, we present PASSION to solve a new challenging task, namely incomplete multi-modal medical image segmentation with imbalanced missing rates. PASSION adopts pixel-wise and semantic-wise self-distillation to regularize modality-specific optimization objectives and preference-aware regularization in task-wise and gradient-wise to balance convergence rates of different modalities. Comprehensive evaluation demonstrates that PASSION outperforms SOTA modality-balancing approaches and works as a plug-and-play module for consistent performance improvement across various backbones.

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
