# OpenReview forum: "PASSION: Towards Effective Incomplete Multi-Modal Medical Image Segmentation with Imbalanced Missing Rates"
_acmmm.org/ACMMM/2024/Conference — MM2024 Oral_

### Official Review · Reviewer_X5B4 · 2024-05-19

**Rating:** 5
**Confidence:** 4

**Summary:**

The author designed Preference-Aware Self-diStillatION (PASSION) for incomplete modality tasks set under imperfect data training. PASSION uses the prediction branch of the fusion as a teacher to transfer knowledge to the single modality feature extraction branches at both the pixel level and the semantic-wise level. To balance learning between modalities, task-level and gradient-level regularization were introduced.

**Strengths:**

The approach proposed in the paper is solving a problem of significant importance. They have proposed a novel approach and demonstrated state-of-the-art results on multiple defined datasets. They have described their approach, datasets, implementation details, evaluation metrics and results very well.

**Limitations:**

1. This is an initial draft and is not yet complete. Figure 2 is difficult to comprehend due to the presence of numerous question marks.
2. Although, I am a bit unsure how this paper is directly related to multimedia but the approach discussed in the paper can certainly be extended to multimedia use cases as well and hence can act as a source of inspiration which can drive further research in the field of multimedia.

**Suitability:**

2

---

### Official Review · Reviewer_YYKA · 2024-05-24

**Rating:** 4
**Confidence:** 4

**Summary:**

This paper introduces a novel approach to address the challenge of incomplete multi-modal medical image segmentation in the presence of imbalanced missing rates. Through Preference-Aware Self-diStillatION (PASSION), the authors propose a method that constructs both pixel-wise and semantic-wise self-distillation mechanisms. This approach aims to balance the optimization objectives across different modalities, enhancing segmentation performance. The paper presents a comprehensive analysis of the proposed method's effectiveness, highlighting its potential to significantly improve segmentation accuracy in challenging scenarios.

**Strengths:**

The paper addresses a more realistic and complex issue in the field of medical image segmentation, specifically incomplete multi-modality segmentation under imbalanced missing rates. This represents a significant advancement as it mirrors the practical conditions encountered in real-world clinical settings where data from different imaging modalities can be incomplete or unequally available.

The proposed method introduces a sophisticated self-distillation framework that operates at both pixel-wise and semantic-wise levels. The solution adjusts the optimization paces of different modalities, ensuring that each modality contributes appropriately to the overall segmentation task. This balancing act enhances the model's ability to effectively utilize available data, leading to improved segmentation performance even when faced with significant modality imbalances during training.

**Limitations:**

The proposed multi-unimodality distillation method bears a resemblance to the techniques presented in [1]. The paper does not adequately differentiate itself from this prior work, leading to an over-claim of its novelty. To strengthen its contribution, the authors should provide a clear comparison with this paper, highlighting the unique aspects and improvements of their approach.

The method described in the paper uses a simple concatenation of different modalities before passing them to the fusion decoder. This approach is relatively basic compared to more advanced fusion techniques utilized in RFNet and mmFormer, or even the simple mean fusion methods employed in [1] and HeMIS. The paper does not provide a rationale for why more sophisticated fusion methods were not considered or implemented. Given that advanced fusion techniques could potentially yield better segmentation performance, the authors need to justify their choice and discuss the limitations of their current fusion strategy.

References:
[1] Q. He, N. Summerfield, M. Dong, and C. Glide-Hurst, "Modality-Agnostic Learning for Medical Image Segmentation Using Multi-modality Self-distillation," *arXiv preprint arXiv:2306.03730*, 2023.

**Suitability:**

3

---

### Official Review · Reviewer_Jq3N · 2024-05-24

**Rating:** 4
**Confidence:** 4

**Summary:**

This paper presents a novel approach, PASSION (Preference-Aware Self-diStillatION), for handling incomplete multi-modal medical image segmentation with imbalanced missing rates. The authors identify and address the realistic scenario where different imaging modalities have varying degrees of availability during model training. PASSION leverages self-distillation to balance optimization objectives across modalities and introduces task-wise and gradient-wise regularization to ensure balanced convergence rates. Experimental results on two datasets demonstrate the effectiveness of PASSION, which also functions as a plug-and-play module compatible with various backbones.

**Strengths:**

(1) Innovative Methodology: The paper introduces PASSION, a unique self-distillation method that balances optimization objectives and convergence rates for different modalities, addressing a significant gap in current research.

(2) Empirical Validation: The experimental results on BraTS2020 and MyoPS2020 datasets show that PASSION outperforms state-of-the-art methods in handling imbalanced modality availability, proving its practical utility in author's own settings.

(3) Plug-and-Play Module: PASSION is designed to be easily integrated into existing models, providing consistent performance improvements across different network architectures.

**Limitations:**

(1) Task settings:

According to Table 3, there seems to be no significant performance difference (approximately 10%) between the IDT and PDT settings. Can the authors further explain the necessity of this task setup?

(2) Limited Evaluation:

Although the experimental results are promising, the evaluation is limited to two datasets. Additional datasets, such as other BraTS datasets, could further validate the generalizability of PASSION. Moreover, it is recommended to include more method comparisons to verify the validity of the proposed method. In Table 3, the results show that the proposed method is effective in this setting, but it is unclear to what extent it is effective. It would be beneficial to add performance comparisons with other algorithms, such as the one mentioned in reference [30].

(3) Parameter Sensitivity:

The paper does not discuss the sensitivity of PASSION to different hyperparameters, which is crucial for understanding its robustness and adaptability. For instance, the choice of missing rates in BraTS2020 (MR = 0.2, 0.4, 0.6, 0.8) needs further explanation. How were these parameters determined, and is there experimental data to justify these specific missing rates? Additionally, is the proposed method effective only with these parameters? It would be beneficial if the authors could add experiments to verify the sensitivity of the method to various hyperparameters.

(4) Suggested Citations:

  [1] Wang, Z., Hong, Y. (2023). A2FSeg: Adaptive Multi-modal Fusion Network for Medical Image Segmentation. In: Greenspan, H., et al. Medical Image Computing and Computer Assisted Intervention – MICCAI 2023.

  [2] Yansheng Qiu, Ziyuan Zhao, Hongdou Yao, Delin Chen, and Zheng Wang. 2023. Modal-aware Visual Prompting for Incomplete Multi-modal Brain Tumor Segmentation. In Proceedings of the 31st ACM International Conference on Multimedia (MM '23)

**Suitability:**

3

---

### Official Review · Reviewer_YoPD · 2024-05-24

**Rating:** 4
**Confidence:** 1

**Summary:**

A new self-distillation solution called PASSION has been proposed to balance different patterns in optimization objectives and pace. It has been proven on two datasets to serve as a plug-and-play module that enhances performance on various backbones.

**Strengths:**

1. For the first time, a new scenario of imperfect data training has been proposed, exploring incomplete multimodal medical image segmentation with imbalanced missing rates.

2. The paper provides a clear description of the problem and detailed methodological details.

**Limitations:**

1. What does the superscript 's' on all symbols after line 381 mean?

2. Do all modalities need to be pre-aligned? For multimodal segmentation tasks, registration is an important prerequisite step; what impact does this have on the applicability of this method?

**Suitability:**

2

---

### Meta-Review · Area_Chair_9jt9 · 2024-07-02

**Recommendation:** Accept (Oral)
**Confidence:** 5

**Metareview:**

There is a consistent recommendation of acceptance from all reviews.